# The Role of *TSHR, PTEN* and *RASSF1A* Promoters’ Methylation Status for Non-Invasive Detection of Papillary Thyroid Carcinoma

**DOI:** 10.3390/jcm11164917

**Published:** 2022-08-21

**Authors:** Raimonda Klimaitė, Mintautė Kazokaitė, Aistė Kondrotienė, Dalia Daukšienė, Rasa Sabaliauskaitė, Kristina Žukauskaitė, Birutė Žilaitienė, Sonata Jarmalaitė, Albertas Daukša

**Affiliations:** 1Institute of Endocrinology, Medical Academy, Lithuanian University of Health Sciences, 50009 Kaunas, Lithuania; 2Laboratory of Genetic Diagnostic, National Cancer Institute, 50009 Vilnius, Lithuania; 3Life Science Center, Institute of Biosciences, Vilnius University, 10257 Vilnius, Lithuania; 4Institute of Digestive Research, Faculty of Medicine, Medical Academy, Lithuanian University of Health Sciences, 50161 Kaunas, Lithuania

**Keywords:** papillary thyroid carcinoma, PTC, methylation, *TSHR*, *PTEN*, *RASSF1A*

## Abstract

Aim: We investigated whether a difference exists between *TSHR, PTEN* and *RASSF1A* methylation status in plasma of subjects with papillary thyroid cancer (PTC). Methods: Peripheral blood samples were collected from 68 patients with PTC and 86 healthy controls (HC). Thyroid cancer tissue and corresponding adjacent normal tissue methylation levels were analyzed. DNA methylation level changes in *TSHR, PTEN* and *RASSF1A* genes were analyzed by quantitative methylation-sensitive polymerase chain reaction. Results: We observed that the methylation level of *TSHR* was significantly higher in the thyroid cancer tissue compared to adjacent normal tissue (*p* = 0.040). *TSHR* methylation levels in the PTC group plasma samples were significantly higher compared to HC (*p* = 0.022). After surgery, PTC plasma samples showed lower *TSHR* and *PTEN* methylation levels compared to the levels before surgery (*p* = 0.003, *p* = 0.031, respectively). The *TSHR* methylation level was significantly higher in PTC with larger tumor size (>2 cm) (*p* < 0.001), and lymph node metastases (*p* = 0.01), lymphovascular invasion (*p* = 0.02) and multifocality (*p* = 0.013) 0ROC analysis revealed that the *TSHR* methylation level provides high accuracy in distinguishing PTC from HC (*p* = 0.022, AUC of 0.616). Conclusion: *TSHR* methylation in peripheral blood samples is expected to be a sensitive and specific minimally invasive tool for the diagnosis of PTC, especially in combination with other diagnostic means.

## 1. Introduction

Thyroid cancer is the most frequent endocrine neoplasm. Based on global cancer statistics in 2020, 586,202 new cases of thyroid cancer worldwide were diagnosed. Papillary thyroid carcinoma (PTC) is the most common thyroid cancer type, accounting for approximately 80–85% of all cases and predominantly occurring in middle-aged adults (45–54 years old) [1]. It carries the best overall prognosis. The 10-year survival rate in PTC patients after indicated treatment is approximately 90% [2,3]. However, approximately 10% of cases may present with metastatic disease at diagnosis/initial presentation, and local or regional PTC recurrences occur in up to 30% of patients [4,5].

Fine needle aspiration biopsy (FNAB) cytology by ultrasonography is still the most widely used method for diagnosing PTC, but it is improbable in up to 15–30% of cases [6,7]. Over the past few decades, a minimally invasive diagnostic test that can accurately diagnose the onset and prognosis of PTC has been the subject of research. DNA methylation status changes have been shown to be related to gene expression alterations. The methylation of the gene promoter is often found in cancer-specific genes and can therefore be used as a cancer marker [8].

Thyroid-stimulating hormone receptor (*TSHR*) is a specific protein on the surface of the thyroid cell membranes, encoded on chromosome 14q31.1 by a gene containing 12 exons. The receptors play an important role in the control of thyroid function and in the pathogenesis of thyroid diseases [9]. Epigenetic research has proven *TSHR*’s role in thyroid carcinogenesis. *TSHR* gene promoter is relatively rich in CpG dinucleotides, and it is believed that the changes in DNA methylation determine *TSHR* gene silencing in thyroid tumors [8]. Activation of the signaling cascades through *TSHR* is the pathway for carcinogenesis and a tumor growth promoter for thyroid cancer. TSH stimulates cytokine production in tumor cells, which could affect the tumor microenvironment [10]. In addition, TSH binds to extrathyroidal cells, including fibroblasts or endothelial cells, and may have the potential to directly modulate the tumor microenvironment. The methylation of the gene promoter is often found in tumor-specific genes and can therefore be used as a cancer marker [11,12].

Phosphatase and tensin homolog gene (*PTEN*) is a tumor suppressor gene located in the 10q23.31 chromosomal region. *PTEN* methylation is generally reported to be closely related to genetic alterations of the *PI3K–AKT* pathway in thyroid tumors. Changes in the *PI3K–AKT* signaling pathway may affect normal thyroid cell growth and proliferation, activate tumorigenesis and indicate thyroid cancer progression [13]. Aberrant promoter silencing of *PTEN* was detected in PTC, follicular thyroid carcinoma and anaplastic thyroid carcinoma cases. Therefore, previous studies have shown that *PTEN* methylation frequency is higher than mutations and loss of heterozygosity in thyroid neoplasms [14].

Ras association domain family 1A (*RASSF1A*) is located on chromosome 3p21.3 and plays an important role in the *Ras/PI3K/AKT, Ras/RAF/MEK/ERK* and Hippo signaling pathways. *RASSF1A* can be inactivated by hypermethylation of its promoter in 20–32% of PTC [15,16]. *RASSF1A* is one of the most common epigenetically inactivated tumor suppressor genes in human cancers. Gene inactivation is caused by altered promoter methylation that can lead to loss of expression of *RASSF1A*. These changes can increase the risk of lung cancer, breast cancer, prostate cancer, ovarian cancer, colorectal cancer, hepatocellular carcinoma, and gastric cancer. Moreover, *RASSF1A* promoter methylation may be a significant prognostic factor for many human cancers, but the relationship between *RASSF1A* promoter methylation changes, and disease pathogenesis is still controversial [17].

The aim of the present study was to determine whether a difference exists between selected genes’ (*TSHR, PTEN*, and *RASSF1A*) methylation status in plasma of subjects with PTC before and after surgery. Moreover, we correlated the methylation levels of selected genes with clinicopathological features and analyzed possible diagnostic value for PTC.

## 2. Materials and Methods

### 2.1. Study Group

Patients with PTC and healthy controls (HC) were involved in our study. Tissue samples were obtained from patients diagnosed with PTC and treated at the Hospital of Lithuanian University of Health Sciences Kaunas Clinics between 2020 and 2022. Plasma samples were obtained from patients with PTC, before and 4–6 weeks after surgery. Surgically resected thyroid tumor tissue and adjacent normal tissue were collected during the surgery. All surgically removed thyroid tissue samples underwent histological examination. The histopathological diagnosis of PTC was confirmed after surgery. Classification of patients with PTC was performed according to the 8th edition of the AJCC/UICC staging system [18]. Histopathological PTC was divided into aggressive (diffuse sclerosing variant and tall cell carcinoma) and non-aggressive (classical and follicular variant) subtypes. The PTC group had no benign nodes of the thyroid and no previous history of any other cancer.

The HC group had no thyroid disease, autoimmune illness, or previous history of any cancer. Thyroid ultrasound, hormone (TSH and fT4) assessments, and anti-thyroid peroxidase (anti-TPO) antibody assessments were performed on all prospective subjects before inclusion in the study.

Plasma thyroglobulin (Tg) was performed on all PTC patients 12 weeks after surgery, before radioiodine therapy. A Tg value of <0.1 ng/mL was considered suppressed.

The study was approved by the Kaunas Regional Committee of Biomedical Research (Lithuania, approval No. BE-2-64; 7 February 2020). Written informed consent was obtained from each participant in the study after a full explanation of the purpose and nature of all procedures used. This study was conducted in accordance with the Declaration of Helsinki.

### 2.2. DNA Samples

Venous blood was drawn from PTC and HC patients. All peripheral venous blood samples (10 mL) were collected in EDTA (BD Vacutainer PPT™ Plasma Preparation Tube; 13 × 100 mm/5 mL) tubes and separated by the ugation 1900× *g* for 10 min at 4 °C. Supernatant was then transferred to a new 15 mL conical tube and centrifuged at 16,000× *g* for 10 min at 4 °C. Purified plasma was transferred to 1.5 mL aliquots and stored at −80 °C until nucleic acid purification.

Thyroid cancer tissue and corresponding adjacent normal tissue were snap-frozen and stored at −80 °C in liquid nitrogen before DNA extraction. Corresponding adjacent normal tissue was removed at least 5 mm away from the primary tumor. Afterward, an experienced pathologist reviewed tissue samples of thyroid cancer tissues consisting of at least 80% cancer cells. Cancer cells were absent in adjacent normal tissue.

### 2.3. DNA Extraction

Plasma cfDNA was extracted from 5 mL blood plasma using QIAamp Circulating Nucleic Acid Kit (Qiagen, Hildigen, Germany), according to the manufacturer’s protocol. Eluted cfDNA was transferred into 0.2 mL Eppendorf tubes and stored at −80 °C.

Genomic DNA was extracted from 25 to 40 mg of frozen tissues using the All Prep DNA/RNA Kit (Qiagen, Hildigen, Germany), according to the manufacturer’s recommendation. DNA concentration was measured using NanoDrop1000 (Thermo Scientific, Waltham, MA, USA).

### 2.4. Bisulfite Conversion

For quantitative methylation-specific PCR (QMSP) analysis, up to 400 ng of purified DNA was modified using the EZ DNA Methylation™ Kit (Zymo Research, Irvine, CA, USA), according to the manufacturer’s protocol; the samples were incubated at 42 °C for 15 min.

### 2.5. Quantitative Methylation-Specific PCR

Target-specific QMSP primers and hydrolysis probes were selected from the previous studies (Appendix A) [19,20] and ordered from Metabion (Martinsried, Germany). In each assay, ACTB was included and was used for normalization [21]. The QMSP was performed in duplicates (analysis of tissue samples) or triplicates (analysis of blood plasma samples) for each set of primers in separate wells. The 20 μl reaction mix consisted of 1× TaqMan^®^ Universal Master Mix II, no UNG (Applied Biosystems™, Waltham, MA, USA), 300 nM of each primer, 50 nM of the probe, and 10 ng bisulfite-converted DNA. All assays were performed under the following conditions: 95 °C for 10 min followed by 45–50 cycles of 95 °C for 15 s and 60 °C for 1 min, using the QuantStudio 5 Real-Time PCR System (Applied Biosystems™, Waltham, MA, USA). A run was considered valid when methylated controls provided a positive signal, and the NTC gave no amplification product. The methylation level of a particular target was estimated based on the ΔΔCq algorithm and expressed as a percentage of the methylation-positive control.

### 2.6. Statistical Analysis

Analysis of the Mann–Whitney U test and Kruskal–Wallis H test criteria for abnormal distribution was used to determine the differences in quantitative traits between the comparison groups. The association between qualitative values in comparative groups was assessed by the chi-square (χ^2^) test. The relationships between the DNA methylation level in the plasma and quantitative parameters were determined by Pearson’s correlation. The predictive capability (diagnostic performance) of each biomarker was investigated by means of the area under the ROC (receiver operating characteristic) curve (AUC). Statistical analyses were performed using SPSS 22.0 software (SPSS Inc., Chicago, IL, USA). The results were considered statistically significant at *p* < 0.05.

## 3. Results

### 3.1. Study Population

Demographic and clinicopathological characteristics of the study population with PTC and HC are shown in Table 1.

A total of 154 patients were included in the study: 68 patients with a histologically confirmed diagnosis of PTC after surgical treatment and 86 healthy controls. Cases and controls were matched for gender and age. The majority of patients were diagnosed with pT1a (39.7%), tumor size ≤ 2 cm (70.6%), the classical variant (42.6%), and with lymphovascular invasion (52.9%) PTC. Table 1. Characteristics of the population with papillary thyroid cancer (PTC) and healthy controls (HC).

### 3.2. DNA Methylation in Thyroid Cancer Tissue and Adjacent Normal Tissue Groups

We examined methylation of *TSHR, PTEN*, and *RASSF1A* in 20 thyroid cancer tissue samples and compared it to adjacent normal tissue. The methylation level of *TSHR* was significantly higher in the thyroid cancer tissue compared to adjacent normal tissue (23.61 vs. 13.62%, *p* = 0.040), while there was no significant difference in *PTEN* and *RASSF1A* methylation between these groups (Figure 1).

### 3.3. DNA Methylation in Plasma from PTC and HC Groups

The methylation level of TSHR was significantly higher in plasma of the PTC patients compared to HC (35.25% vs. 28.57%, *p* = 0.022), while there was no significant difference in PTEN and RASSF1A methylation between these groups (Figure 2a–c).

### 3.4. DNA Methylation in Plasma PTC Patients before and after Surgery

Plasma *TSHR, PTEN*, and *RASSF1A* methylation were compared before and after thyroid surgery. After surgery, samples showed significantly lower levels of *TSHR* and *PTEN* methylation than before surgery. Moreover, the methylation level of *TSHR* and *PTEN* after surgery significantly decreased in PTC patients with suppressed Tg concentrations (*p* < 0.001 and *p* = 0.038, respectively) (Figure 3b). However, after surgery, the level of *RASSF1A* methylation decrease was not statistically significant (Table 2).

### 3.5. Association of DNA Methylation Level in Plasma with Clinicopathological Features of PTC

We analyzed the association of *TSHR, PTEN*, and *RASSF1A* methylation levels with demographic and clinicopathological characteristics of the study population. The *TSHR* methylation level was significantly higher in PTC with larger tumor size (>2 cm) compared to smaller (≤2 cm) tumor size (*p* < 0.001). Patients with lymph node metastases, lymphovascular invasion and multifocality had significantly higher methylation levels of *TSHR* (*p* = 0.010, *p* = 0.020 and *p* = 0.013, respectively). Meanwhile, no statistically significant relationship was found in extrathyroidal extension groups (*p* = 0.621). The methylation levels of *TSHR, PTEN*, and *RASSF1A* in aggressive histology variants subtypes of PTC to other non-aggressive subtypes of PTC were compared, but no statistically significant relationship was found (Table 3).

The total tumor size was calculated as the sum of the diameters of all tumors in PTC multifocal cases. The analysis showed a moderate positive correlation between the methylation level of *TSHR* with the total size of PTC tumors (*p* = 0.009, r = 0.315) (Figure 4).

### 3.6. The Diagnostic Value of Plasma TSHR, PTEN, and RASSF1A Methylation

To evaluate the possible diagnostic value of *TSHR, PTEN*, and *RASSF1A*, ROC analysis was performed. The methylation level of *TSHR* was statistically significant and satisfactory to differentiate PTC patients from HC (*p* = 0.022). *TSHR* had the highest AUC of 0.616 (95% CI = 0.519–0.714), with 83.8% sensitivity and 71.0% specificity. This might be useful for differentiating PTC from HC, especially in combination with other PTC-specific biomarkers. The methylation of *PTEN* and *RASSF1A* did not show statistically significant differences between PTC and HC (Figure 5).

## 4. Discussion

The methylation of DNA promoter is an important and well-known mechanism for carcinogenesis reported in many types of cancers. The promoter CpG islands of genes in healthy cells are generally protected from hypermethylation, but this protection may be lost early in carcinogenesis. This has led to the use of methylation of tumor suppressor genes as biomarkers for the early diagnosis of cancers [22].

The aim of our study was to evaluate the level of promoter methylation of a set of three independent genes (*TSHR, PTEN* and *RASSF1A*) and to assess their diagnostic and prognostic values in papillary thyroid tumors. We examined snap-frozen thyroid cancer and adjacent normal tissue samples. The methylation level of *TSHR* was significantly higher in the thyroid cancer tissue compared to adjacent normal tissue, while there was no significant difference in *PTEN* and *RASSF1A* methylation between these groups. The methylation levels of *TSHR, PTEN*, and *RASSF1A* promoters were compared in peripheral blood samples of PTC and HC. We observed that *TSHR* methylation levels in the PTC group were significantly higher compared to HC, while the levels of *PTEN* and *RASSF1A* methylation did not differ significantly. After surgery, PTC plasma samples showed lower *TSHR* and *PTEN* methylation levels compared to the levels before surgery. The *TSHR* methylation level was significantly higher in PTC with larger tumor size (>2 cm) compared to a smaller tumor size (≤2 cm) and in patients with lymph node metastases, lymphovascular invasion and multifocality. The methylation level of *TSHR* correlated with total tumor size. ROC analysis revealed that *TSHR* methylation level provides higher accuracy in distinguishing PTC from HC and may be used as a potential tumor marker. The specificity, sensitivity, and accuracy of *PTEN* and *RASSF1A* in the diagnosis of PTC showed no significant value.

A growing body of research shows that blood plasma DNA methylation status is associated with cancer in several tissues, which is why methylation status in peripheral blood may be a potential source of non-invasive cancer biomarkers.

There are many studies of DNA methylation analysis in peripheral blood samples for detecting hepatic, colon, lung, breast, stomach, and endometrial tumors [23,24,25]. Meanwhile, there is only one study analyzing global DNA methylation in peripheral blood from PTC patients and control individuals, without significant differences detected [26]. Feng Wei et al. analyzed the methylation status of *PTEN* and *DAPK* in blood and tissue samples of patients with thyroid cancer [27].

Our study shows that *TSHR* methylation levels in the PTC group were significantly higher compared to HC. Moreover, the methylation level of *TSHR* was significantly higher in the thyroid cancer tissue compared to adjacent normal tissue. ROC curve analysis confirmed that the plasma *TSHR* methylation level might be quite a reliable biomarker in discriminating PTC from HC (AUC of 0.616 (95% CI = 0.519–0.714)). J. K. Stephen et al. also demonstrated that methylation of *TSHR* distinguished PTC from normal thyroid tissue [28]. This finding may support the methylation of *TSHR* as a promising biomarker in differentiating PTC patients from healthy individuals, especially when used in combination with other PCT-specific biomarkers.

In most PTC patients with suppressed Tg, the *TSHR* and *PTEN* methylation levels were significantly decreased after surgery. The decrease in Tg levels may take approximately 1 year. In our study, we observed significant differences only 3 months after surgery. Our findings suggest that the plasma methylation levels of *TSHR* and *PTEN* changed significantly after surgery and therefore might be indicators of radical removal of the tumor and useful prognostic markers after thyroidectomy. However, it would be useful to assess the changes after 1 year.

TSHR plays a key role in the regulation of thyroid cell proliferation, differentiation and function. Hypermethylation of TSHR gene, which leads to TSHR expression silencing, plays an important role in the pathogenesis of thyroid cancer. Studies by Wang and Zheng et al. found that the methylation of TSHR gene in PTC patients in later stages was higher than that in earlier stages, which provided evidence that TSHR gene inactivation promotes the progression of PTC. [29,30] Liu T et al. demonstrated that TSHR inhibits metastasis through regulating epithelial-mesenchymal transition (EMT) in vitro, and that a lack of expression of TSHR is a significant independent factor affecting distant metastasis and poor prognosis in DTC [31].

We performed the analysis to explore the relationship between the methylation level of *TSHR* promoter in plasma samples and clinicopathological features of PTC. Previous studies reported that the incidence of *TSHR* promoter methylation in patients with lymph node metastasis is significantly higher than that in PTC with no lymph node metastasis [12]. On the contrary, Dai et al.’s study results showed that the *TSHR* gene methylation rate in patients with no lymph node metastasis was higher than that in patients with lymph node metastasis [32]. In contrast, our results showed that the level of *TSHR* promoter methylation in PTC patients with lymph node metastasis was significantly higher than in patients with no lymph node metastasis. A significant relationship between *TSHR* methylation and tumor diameter was also described in the meta-analysis, as the *TSHR* promoter methylation occurrence in patients with tumor diameter >2 cm was higher than in patients with tumor diameter ≤2 cm [12]. Our findings also suggest that the *TSHR* methylation level was significantly higher in patients with larger tumor size. Moreover, we observed that the *TSHR* methylation level was significantly higher in the presence of lymphovascular invasion and multifocality. On the contrary, Mohammadi-asl J. et al. did not find a significant relationship between *TSHR* methylation and lymphovascular invasion [33]. In Lucieli Ceolin et al.’s study, no significant relationship was observed between DNA methylation levels and tumor size or presence of metastatic disease in thyroid cancer [26].

Feng Wei et al. analyzed the methylation status of *PTEN* in peripheral blood and tissue samples. In both samples, the rate of *PTEN* methylation was significantly higher in PTC patients compared with controls. *PTEN* methylation status was not affected by tumor size but significantly correlated with metastasis in lymph nodes. Moreover, there was no significant difference in the specificity, sensitivity, or accuracy of *PTEN* in the diagnosis of PTC [27]. On the contrary, in our study, there was no significant difference in the *PTEN* methylation level in PTC patients compared with the control group and no relationship between the clinicopathological features of PTC. On the other hand, the plasma methylation level of *PTEN* changed significantly after surgery. Therefore, we support the further investigation of *PTEN* methylation in a large cohort.

The epigenetic value of the *RASSF1A* in thyroid carcinoma has been highlighted. *RASSF1A* methylation can be used as a diagnostic marker in thyroid malignancies [34,35]. On the contrary, in some studies, no significant relationship between *RASSF1A* methylation and thyroid cancer was detected [36,37]. We also did not find any significant relationship between *RASSF1A* methylation and clinicopathological features in PTC and the control group.

Our study has several limitations that restrict *TSHR, PTEN*, and *RASSF1A* methylation to be suggested as markers for PTC detection. Due to a short patient follow-up period after the surgery, it is difficult to determine the prognostic value of these genes’ methylation in PTC. Another limitation is the relatively small sample size.

Altogether, our findings indicate that *TSHR* methylation in peripheral blood samples is expected to be quite a sensitive and specific minimally invasive parameter for the diagnosis of PTC. However, further investigation is required to confirm the diagnostic and prognostic *TSHR* methylation value for PTC. Further insight on epigenetic biomarkers for the non-invasive detection of PTC might be provided by epigenomic techniques in large, independent cohorts.

## Figures and Tables

**Figure 1 jcm-11-04917-f001:**
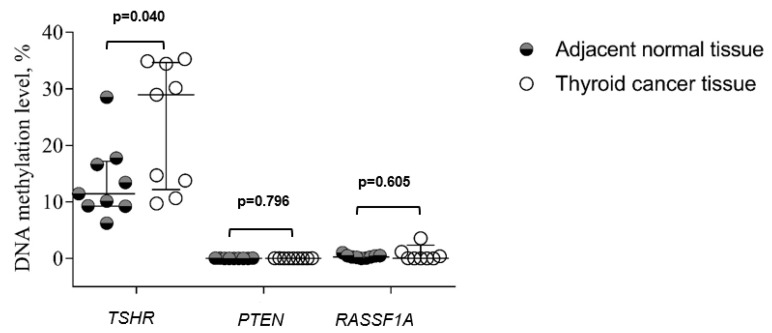
The comparison of *TSHR, PTEN*, and *RASSF1A* methylation levels in thyroid cancer tissue and adjacent normal tissue groups. Analysis of the Kruskal–Wallis test criterion was used to determine the differences in quantitative traits between the comparison groups. Data are expressed in whisker plots for mean and standard deviation (SD).

**Figure 2 jcm-11-04917-f002:**
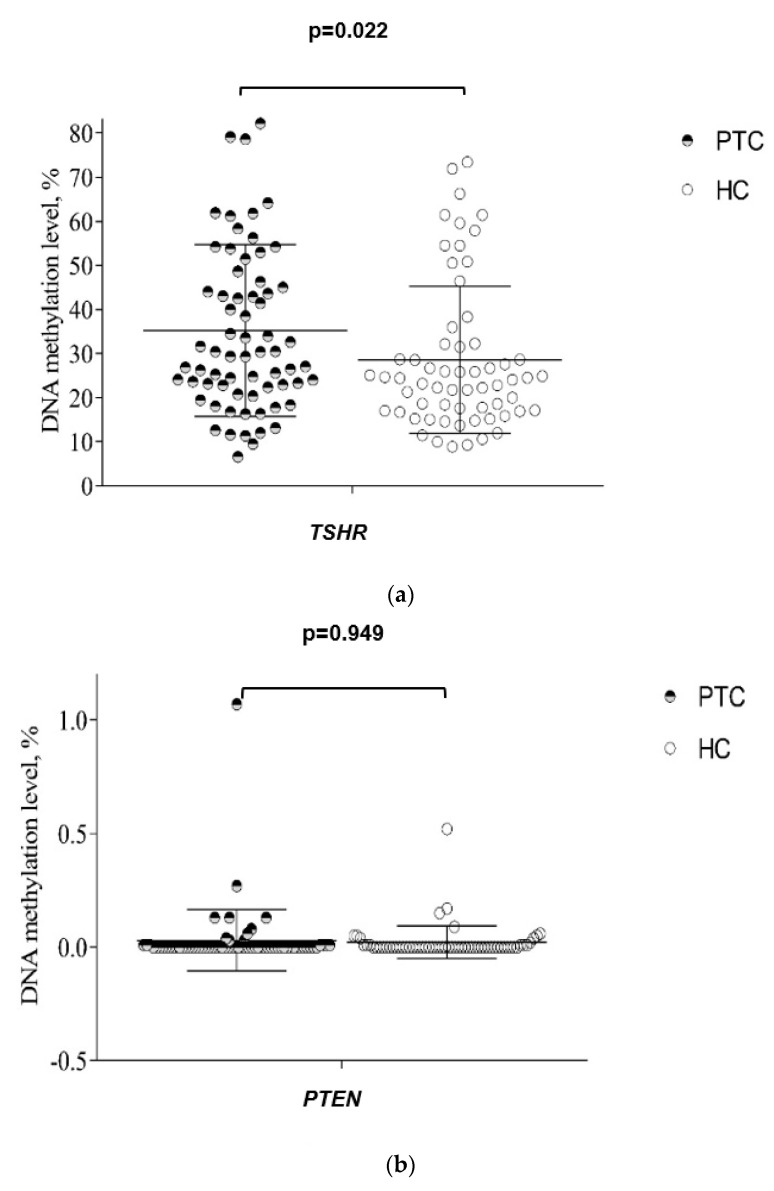
(**a**) The comparison of TSHR methylation level in papillary thyroid cancer (PTC) and healthy control (HC) patients. Analysis of the Mann–Whitney U test criterion was used to determine the differences in quantitative traits between the comparison groups. Data are expressed in whiskers plot for mean and standard deviation (SD). (**b**) The comparison of PTEN methylation level in papillary thyroid cancer (PTC) and healthy control (HC) patients. Analysis of the Mann–Whitney U test criterion was used to determine the differences in quantitative traits between the comparison groups. Data are expressed in whisker plots for mean and standard deviation (SD). (**c**) The comparison of RASSF1A methylation level in papillary thyroid cancer (PTC) and healthy control (HC) patients. Analysis of the Mann–Whitney U test criterion was used to determine the differences in quantitative traits between the comparison groups. Data are expressed in whiskers plot for mean and standard deviation (SD).

**Figure 3 jcm-11-04917-f003:**
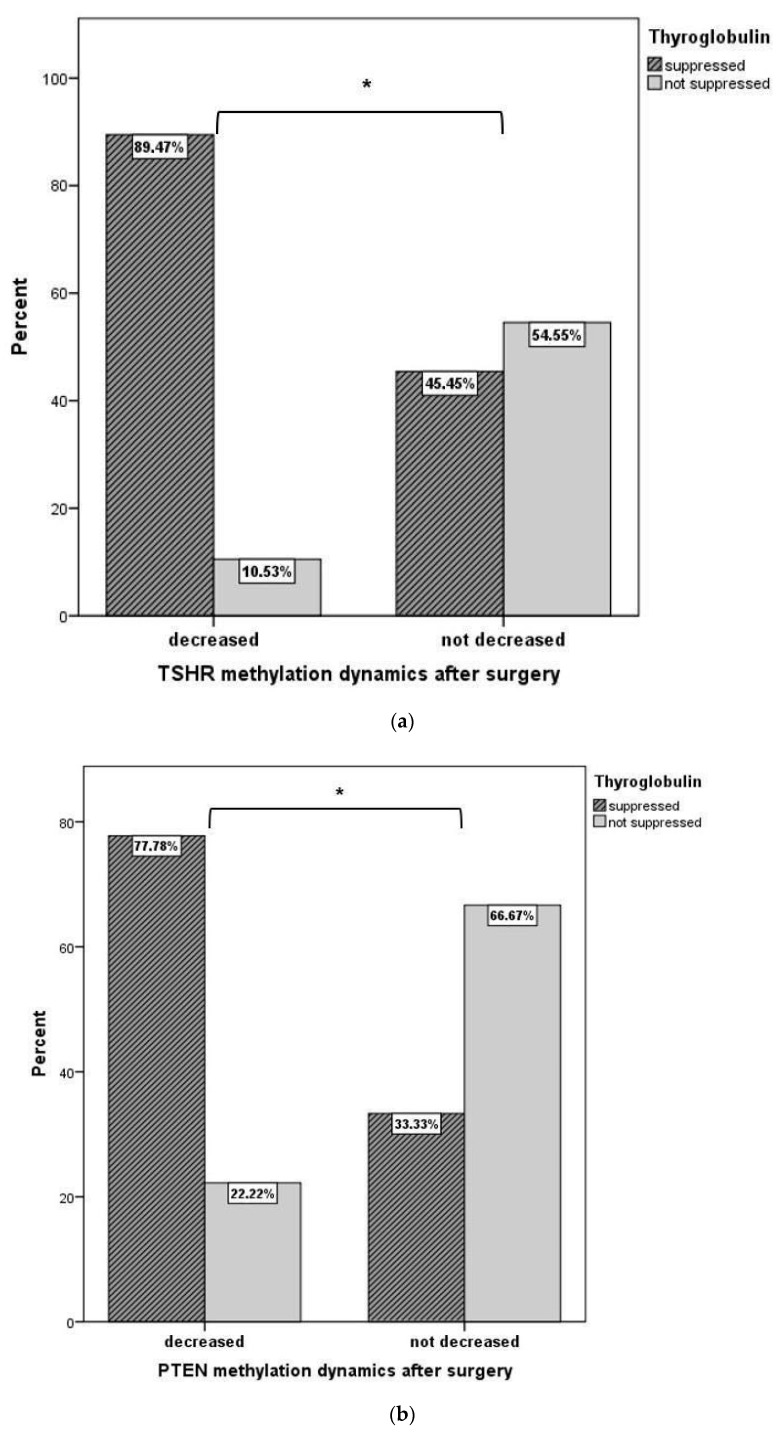
(**a**) The comparison of *TSHR* methylation level dynamics after surgery in PTC patients with suppressed and not suppressed thyroglobulin. * *p* < 0.001. A chi-square independence test was used to determine the differences between the comparison groups. (**b**) The comparison of *PTEN* methylation level dynamics after surgery in PTC patients with suppressed and not suppressed thyroglobulin. * *p* = 0.038. A chi-square independence test was used to determine the differences between the comparison groups.

**Figure 4 jcm-11-04917-f004:**
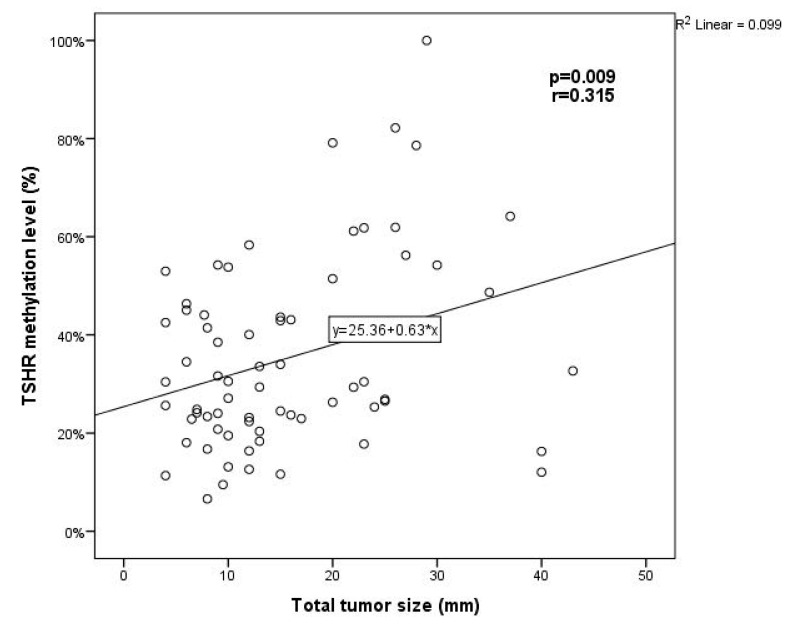
The correlation between *TSHR* methylation level and the total tumor size. The Pearson correlation coefficient was used to measure the strength of a linear association between two variables.

**Figure 5 jcm-11-04917-f005:**
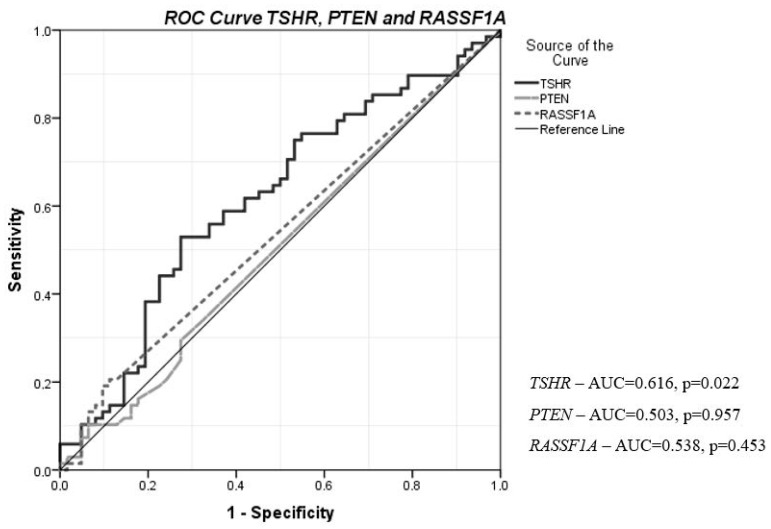
Diagnostic value of plasma *TSHR, PTEN*, and *RASSF1A* methylation levels in discriminating PTC from HC. ROC curves were used to distinguish the groups. ROC, receiver operating characteristic; AUC, area under the curve; PTC, papillary thyroid cancer; HC, healthy control.

**Table 1 jcm-11-04917-t001:** Demographic and clinicopathological characteristics of the study population with PTC and HC.

Characteristic	PTC *n* = 68	HC *n* = 86	*p*-Value
Gender			
Male	8 (11.8%)	11 (12.79%)	
Female	60 (88.2%)	75 (87.2%)	
Age at initial surgery (years)	48.19 (14.9)	45.30 (12.07)	*p* = 0.221
T (TNM), *n* (%)		-	-
pT1a	27 (39.7)		
pT1b	7 (10.3)		
pT2	4 (5.9)		
pT3a	19 (27.9)		
pT3b	11 (16.2)		
Tumor size (cm)		-	-
≤2	48 (70.6)		
>2	20 (29.4)		
Lymph node metastases at initial surgery		-	-
Yes	16 (23.5)		
No	52 (76.5)		
Variant of PTC, *n* (%)		-	-
The classical variant	29 (42.6)		
The follicular variant	18 (26.5)		
The diffuse sclerosing variant	17 (25.0)		
The tall cell carcinoma	4 (5.9)		
Extrathyroidal extension		-	-
Yes	30 (44.1)		
No	38 (55.9)		
Lymphovascular invasion		-	-
Yes	36 (52.9)		
No	32 (47.1)		
Multifocality		-	-
Yes	16 (23.5)		
No	52 (76.5)		

**Table 2 jcm-11-04917-t002:** Plasma *TSHR, PTEN*, and *RASSF1A* methylation in papillary thyroid cancer (PTC) patients before and after thyroidectomy. Analysis of Wilcoxon signed-rank test was used. * *p* < 0.05.

Marker	Methylation Level: MEAN (SD)		*p*-Value
	Pre-Operative PTC (*n* = 68)	Post-Operative PTC (*n* = 62)	
*TSHR*	34.105 (17.790)	26.901 (11.617)	0.003 *
*PTEN*	0.029 (0.134)	0.003 (0.014)	0.031 *
*RASSF1A*	1.010 (3.248)	0.816 (4.072)	0.903

**Table 3 jcm-11-04917-t003:** Correlation between clinicopathological features of papillary thyroid cancer (PTC) and methylation level of *TSHR, PTEN*, and *RASSF1A* in plasma samples. * *p* < 0.05.

	Methylation Level: MEAN (Minimum–Maximum)		
Characteristic	TSHR	PTEN	RASSF1A
	35.24 (6.60–100.00)	0.02 (0.00–1.07)	1.18 (0.00–36.74)
	%	%	%
	*p*-Value	*p*-Value	*p*-Value
*p*-ValueGender	0.505	0.456	0.059
Male	33.06 (11.35–78.60)	0.01 (0.00–0.06)	0.94 (0.00–6.66)
Female	35.54 (6.60–100.00)	0.03 (0.00–1.07)	1.02 (0.00–21.74)
Age at initial surgery (years)	0.599	0.204	0.061
≤55 years	35.95 (11.63–100.00)	0.04 (0.00–1.07)	0.79 (0.00–21.74)
>55 years	33.95 (6.60–79.13)	0.01 (0.00–0.13)	1.41 (0.00–9.36)
pT (TNM)	0.422	0.148	0.627
pT1	37.15 (6.60–100.00)	0.04 (0.00–1.07)	1.35 (0.00–21.74)
pT2–3	33.22 (11.35–79.13)	0.01 (0.00–0.13)	0.65 (0.00–7.10)
Tumor size (cm)	<0.001 *	0.743	0.164
≤ 2	28.84 (6.60–58.32)	0.04 (0.00–1.07)	1.28 (0.00–21.74)
> 2	50.62 (12.04–100.00)	0.01 (0.00–0.13)	0.36 (0.00–7.22)
Lymph node metastases at initial surgery	0.010 *	0.368	0.487
Yes	47.01 (12.62–82.19)	0.04 (0.00–0.27)	1.04 (0.00–9.36)
No	31.62 (6.60–100.00)	0.03 (0.00–1.07)	1.00 (0.00–21.74)
Variant of PTC	0.300	0.791	0.824
Aggressive histology of PTC	31.32 (12.62–64.15)	0.01 (0.00–0.08)	0.95 (0.00–9.36)
Non-aggressive subtypes of PTC	37.12 (6.60–100.00)	0.04 (0.00–1.07)	1.04 (0.00–21.74)
Extrathyroidal extension	0.621	0.608	0.501
Yes	38.12 (13.13–100.00)	0.05 (0.00–1.07)	0.65 (0.00–7.22)
No	32.97 (6.60–78.60)	0.01 (0.00–0.27)	1.24 (0.00–21.74)
Lymphovascular invasion	0.020 *	0.726	0.578
Yes	40.96 (12.62–100.00)	0.04 (0.00–1.07)	0.72 (0.00–7.22)
No	28.81 (6.60–61.79)	0.02 (0.00–0.27)	0.42 (0.00–21.74)
Multifocality	0.013 *	0.094	0.437
Yes	47.19 (13.13–100.00)	0.08 (0.00–0.27)	0.86 (0.00–7.10)
No	31.57 (6.60–82.19)	0.01 (0.00–1.07)	1.06 (0.00–21.74)

## Data Availability

Not applicable.

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
