# Peer review of "The Role of TSHR, PTEN and RASSF1A Promoters’ Methylation Status for Non-Invasive Detection of Papillary Thyroid Carcinoma"

_jcm, 2022, doi:10.3390/jcm11164917_

Round 1

Reviewer 1 Report

Comments to the Author:

The study data are valuable. I suggest to publish it with minor changes indicated below.

Material and methods:

The “study group” should be the first chapter.

Page 3, line 123: “Bisulfite conversion” is standing alone.

Page 3, line 142-146: The whole paragraph would be better placed on the end of the study group description.

Results:

Page 4, chapter: 3.1. Study population: few sentences to conclude the findings from the table would be welcomed.

Figures 3a and 3b: please explain what is the meaning of the right part of the figures (not decreased bars); from where those percentages come?.

The first part of the Table 3 is not clearly presented, for example lacking the word methylation, …

Author Response

Dear Reviewer,

enclosed with this letter is the revised version of our manuscript “The role of TSHR, PTEN and RASSF1A promoter’s methylation status for non-invasive detection of papillary thyroid carcinoma “, which we would like to resubmit for publication as an article in Journal of Clinical Medicine.

We have taken up the criticism and suggestions of the Reviewer. In the accompanying rebuttal letter are our point-by-point responses to each comment. Please see the attachment.

In the revised manuscript, all changes have been indicated by “Track Changes” function or highlighted. 

The manuscript submitted to MDPI for English editing has been edited. English editing ID: English-48111. 

We expect that the revisions in the manuscript along with the accompanying responses will be considered for publication.

Raimonda KlimaitÄ—

Reviewer 2 Report

In the present work, KlimaitÄ— and colleagues describe the detection of tumor and plasma methylation of specific loci as a possible non-invasive method for the diagnostic evaluation and stratification of patients with papillary thyroid carcinoma.

The work is introduced by a description of the pathological context and a short state of the art of available biomarkers for the diagnostic/prognostic evaluation of patients affected by these neoplasia. The Authors then describe different strategies of detection of the tumor and plasma DNA methylation. They especially focus on the methylation of three loci, TSHR, PTEN and RASSF1A, whose altered methylation patterns might be implied in the carcinogenesis of these tumors. They therefore propose that methylation of these loci represent interesting targets for non-invasive diagnostic/prognostic evaluation of patients affected by these neoplasia.

Overall, the article is well written, even if some typos make some statements unclear (I indicate one possible example as a minor point below).  The introduction provides quite sufficient elements for a non-specialist to understand the relevance of the questions.

Authors used a DNA methylation sensitive real-time PCR method to quantify the extent of DNA methylation in a large cohort of papillary carcinomas and surrounding non tumor tissue, as well as in the blood plasma of patients and healthy individual controls in order to test the correlation between DNA methylation extent of TSHR, PTEN and RASSF1A and clinico-pathological features.

Identification of non-invasive biomarkers for cancer diagnosis and stratification is in general an interesting question. However, presentation of the data is not clear as figure legends do not provide sufficient element to the reader to understand the way data are normalized and/or relativized. Statistical tests that are used should be clearly stated in every figure legends, as well as the dispersion of data (SD, SEM, etc..).

1.     First of all, I am not sure I understand how the methylation profiles of promotor regions of PTEN and RASSF1A loci, appearing to be almost unmethylated or poorly methylated in all the conditions, might be of any interest (see figures 1, 2B, 2C). In particular in the case of PTEN, more than 90% of samples tested show no methylation at all of the considered promoter region. Based on this consideration, the evaluation of PTEN (and RASSF1A) does not deserve in my opinion any particular focus for the rest of the following narrative compared to any other randomly selected gene.

2.     Concerning the methylation of  TSHR, a large proportion of the conclusions made by the authors seem to be based on differences that are barely statistically convincing. For example, methylation extent in peripheral blood of “35.25 (patients) vs 28.57 (controls) %, p=0.022“ described in figure 2A does not seem to me to represent a convincing argument to propose this measure as a diagnostic tool due to the large overlap between the two groups. In figure 5, Authors should clearly indicate the threshold value at which  they observe “83.8% sensitivity and 71.0% specificity“ for this dataset. Looking at the dispersion of data in figure 2A, it is quite hard to believe that DNA methylation of this region would represent such a specific test to avoid considering healthy individual as at risk of having a papillary carcinoma.

3.     The results described in section 3.4 on the dynamics of DNA methylation of these loci before and after surgery are not clear to me. How can the PTEN (and TSHR) methylation extent (in %) be that elevated compared to the results presented in figures 2A and 2B ? Is there any relativization that has been made ?

4.     Section 3.5 on the association between DNA methylation of the three genes and clinic-pathological figures appears to be more convincing that the others and should probably represent the real main focus of the manuscript. However, calculating the tumor size as “sum of the diameters of all tumors in PTC 246 multifocal cases” may be confusing, as increased tumor size might actually simply reflect the duration of the disease rather than its aggressiveness or growth rate. Can the Authors provide any evidence to relate the extent of DNA methylation with the proliferation rate of tumors (e.g. by scoring the proliferation index by immunohistological methods) ? Also, it is not very clear to me why the authors decided to show the results in sections 3.4 and 3.5 as tables rather than plotting the results on graphs.

5.     Can the Authors show some clear examples of IHC used for pathological considerations (e.g. lymphovascular invasion) and quantified data to support their conclusions, and try to explain why hypermethylation of TSHR is associated with a more invasive/metastatic phenotype ?

Overall, I have many doubts that measuring the methylation of these genes really represents an improved strategy for the diagnosis of PTC,  but some interesting conclusions could be made concerning the correlation with clinicopathological progression. This latter point will require more evidences to be presented.

Minor points :

·      At line 73, it is mentioned that “RASSF1A can be inactivated by hypomethylation of its promoter”. Is it true ?

Author Response

Dear Reviewer,

enclosed with this letter is the revised version of our manuscript “The role of TSHR, PTEN and RASSF1A promoter’s methylation status for non-invasive detection of papillary thyroid carcinoma “, which we would like to resubmit for publication as an article in Journal of Clinical Medicine.

We have taken up the criticism and suggestions. In the accompanying rebuttal letter are our point-by-point responses to each comment. Please see the attachment.

In the revised manuscript, all changes have been indicated by “Track Changes” function or highlighted. 

The manuscript submitted to MDPI for English editing has been edited. English editing ID: English-48111. 

We expect that the revisions in the manuscript along with the accompanying responses will be considered for publication.

Raimonda KlimaitÄ—

Reviewer 3 Report

In the manuscript entitled “The role of TSHR, PTEN and RASSF1A promoter’s methylation status for non-invasive detection of papillary thyroid carcinoma”, the authors aim to investigate the TSHR, PTEN and RASSF1A methylation status in plasma of patients with papillary thyroid cancer (PTC). Several critical concerns need to be addressed.

1.      The association between the methylation status of TSHR, PTEN and RASSF1A and PTC has been well studied. Thus, the novelty of the current study is limited.

2.      Why did the authors focus on the methylation status of TSHR, PTEN and RASSF1A in plasma of PTC patients? The reasons should be explained.

3.      In current study, the authors detected the methylation status of TSHR, PTEN and RASSF1A in 68 patients. The sample size should be enlarged.

4.      Ethical approval is common to all research involving human participants. However, the authors did not mention it in the manuscript.

5.      How did the authors calculate the ROC curve? A detailed explanation should be provided.

6.      The limitation of the study should be discussed.

Author Response

(The authors gave the same response as above.)

Round 2

Reviewer 2 Report

In this version, Authors have improved the presentation by following few of the Reviewer's suggestion. Overall, the presentation seems a little more accurate and the introduction is easier to understand.

However, none of the main issues that were raised in my case have really been addressed with any real data, and I think that part of the conclusions overstate the significance of the dataset presented.  As an example, "TSHR had the highest AUC of 0.616, with 83.8% sensitivity and 71% specificity..." (line 291), implies a risk of 29% in assigning healthy individuals to the PTC class, which would be quite problematic for clinicians and do not justify the intent of this section. Authors should stick to the results (p=0.022) and mention that, in the future, the measure of methylation of this region  might somehow  represent a diagnostic/prognostic parameter to be used in combination with other unidentified parameters.

Although I do not doubt the existence of the analyses, it is always quite frustrating for a cancer biologist to accept conclusions made on anatomopathological observations that are not documented by any visual proof or  objectivequantification. Section 3.5 on association of DNA methylation and clinicopathological features of PTC is entirely based on this kind of undocumented correlations that readers will never be able to judge by their own.

Moreover, I still think that figure legends are not very accurate. E.g. Authors should mention whether the bars in figures 1 and 2 represent mean or median values.

Overall, I think that until not supported by further analyses, the significance of this work and its scientific soundness remains very limited but I have no other requests for the Authors that I hope will better clarify some of these issues in the future.

Author Response

Dear reviewer,

We have taken up the criticism and suggestions. In the accompanying rebuttal letter are our point-by-point responses to each comment.

1. However, none of the main issues that were raised in my case have really been addressed with any real data, and I think that part of the conclusions overstate the significance of the dataset presented.  As an example, "TSHR had the highest AUC of 0.616, with 83.8% sensitivity and 71% specificity..." (line 291), implies a risk of 29% in assigning healthy individuals to the PTC class, which would be quite problematic for clinicians and do not justify the intent of this section. Authors should stick to the results (p=0.022) and mention that, in the future, the measure of methylation of this region  might somehow  represent a diagnostic/prognostic parameter to be used in combination with other unidentified parameters.

Response: In agreement with the reviewer comments, we modified the text in the Abstract, Results and Discussion, pointing on the need to increase the specificity of the epigenetic test by using a combination of biomarkers or other diagnostic means.

2. Although I do not doubt the existence of the analyses, it is always quite frustrating for a cancer biologist to accept conclusions made on anatomopathological observations that are not documented by any visual proof or objectivequantification. Section 3.5 on “association of DNA methylation and clinicopathological features of PTC“ is entirely based on this kind of undocumented correlations that readers will never be able to judge by their own.

Response: Clinicopathological evaluation of the tumor is a part of cancer diagnostics performed according to international standards (AJCC Cancer Staging Manual 8 th edition (Doescher J, Veit JA, Hoffmann TK. [The 8th edition of the AJCC Cancer Staging Manual: Updates in otorhinolaryngology, head and neck surgery]. HNO. 2017 Dec;65(12):956-961. DOI: 10.1007/s00106-017-0391-3. PMID: 28717958)), thus pathology, radiology images other clinical measure usually are not provided in scientific papers as a part of research material. We agree with the reviewer that the “total tumor size” is a derivative measure and can be imprecise. An association with this measure was deleted from the Abstract. However, multifocality is common in PTC, with an incidence up to 37% and has been confirmed as a risk factor affecting the aggressiveness and prognosis. The size of multifocal tumors in the current staging system (AJCC 8th edition, Materials and Methods (in lines 97-99)) is still described in the same way as that of unifocal tumors, focusing on the maximum diameter. In recent years, the concept of “total tumor size” has also been reported by many studies and has been demonstrated to be closely related to many tumor characteristics and even prognosis. Therefore, it has been suggested that “total tumor size” should be included in methylation analysis as derivate pathological measure.

3. Moreover, I still think that figure legends are not very accurate. E.g. Authors should mention whether the bars in figures 1 and 2 represent mean or median values.

Response: Correction was made and marked with ‘Track changes’.

In the revised manuscript, all changes have been indicated by “Track Changes” function.

We expect that the revisions in the manuscript along with the accompanying responses will be considered for publication.

Raimonda KlimaitÄ—

Reviewer 3 Report

No further comments.

Author Response

.